# Potential Markers for Selecting Self-Eliminating Apple Genotypes

**DOI:** 10.3390/plants10081612

**Published:** 2021-08-05

**Authors:** Aurelijus Starkus, Birute Frercks, Dalia Gelvonauskiene, Ingrida Mazeikiene, Rytis Rugienius, Vidmantas Bendokas, Vidmantas Stanys

**Affiliations:** Lithuanian Research Centre for Agriculture and Forestry, Department of Orchard Plant Genetics and Biotechnology, Institute of Horticulture, Babtai, LT-54333 Kaunas, Lithuania; aurelijusstarkus@gmail.com (A.S.); dalia.gelvonauskiene@lammc.lt (D.G.); ingrida.mazeikiene@lammc.lt (I.M.); rytis.rugienius@lammc.lt (R.R.); vidmantas.bendokas@lammc.lt (V.B.); vidmantas.stanys@lammc.lt (V.S.)

**Keywords:** apple, carbohydrates, dynamics of fruitlet self-elimination, leaves, plant hormones, seeds

## Abstract

The heavy blooming of apple trees results in the inefficient usage of energy and nutritional material, and additional expenditure on fruitlet thinning is required to maintain fruit quality. A possible solution for controlling the fruit load on trees is the development of new cultivars that self-eliminate excess fruitlets, thus controlling yield. The aim of our study was to identify biological differences in apple cultivars in terms of blooming intensity and fruitlet load self-regulation. In total, 19 apple cultivars were studied in the years 2015–2017. The dynamics of fruitlet self-elimination, seed development in fruitlets and fruits, photosynthetic parameters, carbohydrates, and plant hormones were evaluated. We established that apple cultivars self-eliminating a small number of fruitlets need a lower number of well-developed seeds in fruit, and their number of leaves and area per fruit on a bearing branch are larger, compared to cultivars, self-eliminating large numbers of fruitlets. A higher carbohydrate amount in the leaves may be related to smaller fruitlet self-elimination. The amount of auxin and a high indole-3-acetic acid/zeatin ratio between leaves of cultivar groups with heavy blooming were higher than in cultivars with moderate blooming. A lower amount of abscisic acid was found in heavy-blooming cultivars during drought stress. All these parameters may be used as markers for the selection of different apple genotypes that self-eliminate fruitlets.

## 1. Introduction

Apple (*Malus* × *domestica* Borkh) is one the most important fruit trees in the temperate climate zone. Stable bearing and the production of high-quality fruits are crucial for growers. Only 5–10% of flowers grow to fruits; therefore, heavy blooming results in inefficient usage of energy and nutritional material [1]. To produce a consistent, high-quality harvest and to eliminate tree biennial bearing in industrial orchards, fruitlet and fruit thinning is used; however, this process is costly and ineffective in some cases [2,3]. The efficiency of chemical fruit thinning depends on the plant’s physiological condition and the environment [4]. Additionally, the chemicals used for thinning may negatively impact tree growth, fruit bearing, and the populations of pollinators. Apple bearing is determined by many factors: meteorological conditions, agrotechnical methods, and the biological parameters of a tree. One of the desired traits in future apple cultivars is the genetically determined self-regulation of fruit load. The dropping processes of reproductive organs occur in the region of the somatic body of plant known as the abscission zone (AZ). Abscission has been a useful process in evolution; it results in the development and spreading of only high-quality fruits and seeds. The transport of indole-3-acetic acid (IAA) through the AZ regulates the sensitivity of the AZ to ethylene [5,6]. AZ becomes sensitive to ethylene when the source of IAA is removed, and abscission starts because of the action of cell membrane enzymes [7]. It has been established that carbohydrates are important in fruit drop [8,9,10]. A lack of carbohydrates increases the production of reactive oxygen species and abscisic acid, thus activating ethylene signaling pathways. As a result, fruit abscission is activated as well. Plants use photosynthesis to produce carbohydrates in leaves, which are used in the control of endogenic and environmental signals. However, the role of leaves in fruitlet drop is unclear.

Studies were conducted on the creation of easily handled systems for the advance identification of abscised fruitlets and the planning of technological means for fruitlet load regulation [11]. The authors established that the measurement of fruit growth speed might be used for determining fruitlets that will drop during the June drop [11]. Lee et al. established that frost shock during early fruit development stages results in changes in the expression of abscisic acid synthesis genes and in the activation of AZ movement in the cell cytoplasm [12].

Several studies in model plants identifying genetic and physiological parameters of fruit drop were performed. Possible gene targets for future research and rearrangement using biotechnological methods were proposed [13]. Gene groups responsible for ethylene biosynthesis, auxin carriers, metabolism of carbohydrates, and hydrolytic ferments were established [14]. However, gene expression may be different and specific to various cells and tissues [15]. Knowledge of the physiological, genetic, and genomic aspects related to fruit abscission is sparse. The self-elimination of fruitlets seems to be an important trait, leading to yield stability and lower orchard management costs; therefore, cultivars with this trait are desired by industrial growers. Modern apple breeding programs often include fruitlet self-elimination in breeding schemes.

The aim of our study was to evaluate the biological differences in apple cultivars in terms of fruitlet load self-regulation and identify features closely related to self-eliminating fruitlets. 

## 2. Results

### 2.1. Evaluation of Dynamics of Blossom and Fruitlet Self-Elimination during Vegetation

The apple tree load of blossoms and fruitlet self-elimination depend on the cultivar. Fluctuations in blooming are characterized by the variation coefficient, which ranges from 1.9% to 72.7% (Table 1). Cultivars with heavy blooming belonging to groups I and II had a similar average blossom number per 1 m of branch during a 3-year period, without significant differences. These cultivars had twice as many blossoms compared to cultivars from groups III and IV, which had a moderate bloom. Cultivar groups I and III showed high self-elimination of fruitlets. However, after the first drop, they had a much higher percentage of remaining fruitlets (18.6–50.8% on average) compared to cultivars in groups II and IV, which showed moderate fruitlet self-elimination (7.7% and 20.1% on average, respectively). The fruitlet number on groups I and III trees decreased more than two-fold until July, whereas the number of fruitlets decreased by one-third on trees in groups II and IV. Therefore, the dynamics of fruitlet self-elimination in cultivar groups were different.

### 2.2. Seed Number in Fruitlets 

Cultivars had an uneven number of seeds per fruitlet. The highest number of seeds was observed in fruits of Ottawa and Orlovim cultivars, with 15.5 and 13.6 seeds per fruitlet, respectively. Fruits of the Yellow Arkad cultivar had only 6.6 seeds (data not shown). We established that cultivars in group II had the lowest share of developed seeds (Figure 1), and this share was significantly different from the cultivars from groups III and IV during both first and second fruitlet drops, at 78.0% and 69.5%, respectively. Furthermore, the lowest share of developed seeds was necessary to ensure the good development of fruits in cultivars from group II (63.5–63.9%). The lowest number of developed seeds was found in heavy blooming cultivars from groups I and II during drop (70.8% and 63.5%, respectively) compared to moderate blooming cultivars from groups III and IV, with 83.5% and 86.4% of seeds developing, respectively.

During drop I, the fruitlets of cultivars belonging to different groups had a small proportion of normally developed seeds (Figure 2). The smallest amount of normally developed seeds was found in cultivar group III (1.8%). A significantly higher proportion of normal seeds was recorded after drop II in fruitlets of all cultivars compared to drop I, and differences between groups could be observed. Fruitlets of group III still had the lowest proportion of normal seeds (9.1%), whereas the share of normal seeds reached 26% in the fruitlets of group IV. Fruit trees from groups I and III, which self-eliminate large numbers of fruitlets, had a significantly lower number of normal seeds (10.8% and 9.1%, respectively) than fruitlets from group II and IV cultivars (21.4% and 26.0%, respectively), which self-eliminate a lower number of fruitlets. Therefore, cultivars that self-eliminate large numbers of fruitlets require a higher number of well-developed seeds for successful fruit development.

Correlation coefficients between seed development and fruit survival in the different apple tree cultivar groups were evaluated (Table 2). Significant differences were not found in group I cultivars; significant differences were found in cultivars from groups II and III at drop II, and a strong positive correlation was observed in group IV cultivars during both drops I and II. A significant correlation was established between fruit survival and well-developed seed number in group II at drop II (r = 0.536), group III at drop I (r = 0.669), and group IV at drops I and II (r = 0.995 and 0.801, respectively). The correlation between the number of non-developed seeds and fruitlet survival at drops I and II varied between cultivar groups; the correlation was strongly negative (r = −0.995) in group IV in June. In the fruits of the cultivars from groups I, II, and III, the number of non-developed seeds was inversely correlated with fruit survival in July (average correlation r = −0.654, −0.650, and −0.879, respectively). This may be explained by the faster development of these cultivar leaves by the faster transition from consumers to producers and the changes in plant hormone balance. 

### 2.3. Impact of Leaf Number and Area on Fruit Development

The number of leaves per fruit varied between the groups of cultivars in the first-year shoots from 3.1 (group III) to 20.4 (group II), on average (Figure 3). The lowest number of leaves per fruit was observed in cultivars that tend to self-eliminate a large number of fruitlets, 8.3 in group I and 3.1 in group III. Therefore, cultivars self-eliminating a lower number of fruits, groups II and IV, had a significantly higher number of leaves on first-year shoots per fruit: 20.4 and 15.3, respectively. A similar ratio trend was observed in leaves on fruiting branches, where groups I and III had fewer leaves per fruit, 23.4 and 7.6, respectively, compared to 81.5 and 44.8 in groups II and IV, respectively.

The number of leaves of the first-year shoots (cultivar groups I, II, and IV) correlated with blooming intensity (Table 3), which reflects the biological relationship between the generative and vegetative development of a plant. First-year shoots first consume plant nutritional materials; only after they reach certain development stage do they start producing the nutritional materials necessary for fruits [16].

The number of leaves on first year shoots in June had a positive impact on seed development; however, the correlation was weak and insignificant in cultivars from group I. An average correlation was observed in cultivars from groups II and III (r = 0.666 and 0.766, respectively) and a strong correlation in group IV (r = 0.970). The leaf number on a fruiting branch correlated with blooming. An average correlation was established for cultivars from group I and a strong correlation for cultivars in groups III and IV (r > 0.9). A positive impact of leaf number per fruiting branch on fruit seed development was observed in cultivars from groups I and II (r = 0.796 and 0.891, respectively), whereas a negative correlation between those traits was observed in group IV cultivars (r = −0.728). No correlation in group III cultivars was observed.

Leaf assimilation area was measured both on first year shoots and fruiting branches. We found that, on average, leaf area on the first-year shoot per fruit ranged from 87.5 cm^2^ (group III) to 379.2 cm^2^ (group IV) (Figure 4). Leaf area on the first-year shoot per fruit in cultivar groups I and III ranged from 87.5–211.7 cm^2^ and was significantly lower than in cultivars from groups II and IV (343.8 and 379.2 cm^2^, respectively). Fruiting branches, on average, had double the leaf assimilation area per fruit, which ranged from 138.9 to 773.1 cm^2^; however, the tendency was similar to that observed in first-year shoots. The leaf area on fruiting branches per fruit was significantly lower in groups I and III than in groups II and IV and ranged from 138.9–382.2 to 610.9–773.1 cm^2^, respectively. 

We found that leaf area per fruit on first year shoots negatively correlated with tree blooming and the percentage of remaining fruits (Table 2). The number of developing seeds in cultivars from group II at drop I correlated with the leaf area on first-year shoots. The leaf area on fruiting branches correlated with blooming in cultivar groups I and IV and with the number of remaining fruitlets in cultivar groups I, II, and IV. At drop I, seed development correlated with leaf area on fruiting branches in cultivar groups III and IV (r = 0.999 and 0.861, respectively). At drop II, a strong positive correlation was observed between the leaf area on fruiting branches and seed development in cultivar groups I and III (r = 0.956 and 0.966, respectively), whereas in cultivar groups IV and II, the correlation between those traits was negative (r = −1.000 and −0.373, respectively).

Our findings show that leaf number and area on first-year shoots negatively correlated with yielding components, and positively correlated with seed development in most cultivars. Leaf number and area on fruiting branches correlated with tree blooming and the number of remaining fruits and had a positive impact on the development of seeds in most cultivars at drop I. However, the impact on seed development differed between the studied cultivar groups at drop II. 

### 2.4. Amount of Carbohydrates in Leaves

The total carbohydrate amount decreased in most of the investigated cultivars during fruitlet drop compared to the carbohydrate amount during blooming (Table 4). Significant differences were observed in cultivars belonging to groups I and III, which show high fruitlet self-elimination. A non-significant decrease in total carbohydrate amount was measured in cultivars belonging to groups II and IV. A similar trend was observed in sorbitol amount: a significant decrease in cultivars belonging to groups I and III. McIntosh, Discover, and Hume cultivars were distinguished by an increase in total carbohydrate amount of 9–19%. 

### 2.5. Plant Hormone Dynamics

Amount of auxin (IAA) in leaves varied between cultivars grouped according to their yield self-regulation habits (Table 5) and meteorological conditions. The IAA amount in cultivar groups with heavy blooming (groups I and II) was higher than in cultivars with moderate blooming (groups III and IV) during the three years of the study. The amount of IAA in the leaves of the same trees was similar during each fruitlet drop; however, the tendencies were unclear. In general, the 3-year data on auxin showed that fruit trees with heavy blooming (groups I and II) had a higher amount of IAA than cultivars with moderate blooming (groups III and IV); this tendency was clear at fruitlet drop I. Drought stress may reduce the auxin amount in various cultivars as well.

The amount of cytokinin zeatin (Z) in the leaves of apple trees during fruitlet drop ranged from 6.8 to 73.9 ng/g FW in groups I and II, respectively, and this fluctuation was dependent on the weather conditions. At drop I in 2015, cultivars from groups I and II had a significantly lower amount of zeatin (6.9 and 18.6 ng/g FW, respectively). A similar trend was observed in July. Weather conditions were favorable for fruit growing in 2016 and 2017. A significantly higher amount of zeatin was found in the leaves of cultivars from group I (42.0 ng/g FW) at drop I and group II (73.9 ng/g FW) at drop II in 2016; however, no significant differences emerged in 2017. Significant differences were observed between cultivars grouped according to blooming intensity. In general, the 3-year data on zeatin amount showed that cultivars with moderate blooming have more zeatin than cultivars with heavy blooming.

The amount of gibberellic acid (GA3) in apple leaves ranged from 0.45 to 3.20 µg/g FW (Table 4) during fruitlet drops and in different years. We could not establish a distinct pattern of impact of GA3 on the regulation of fruit tree blooming and self-elimination of fruitlets. However, a tendency of the leaves of trees from group II having a higher GA3 amount was observed.

The amount of abscisic acid (ABA) in leaves was low during the entire study period, but different trends were observed each year. The amount of ABA ranged from 10.6 ng/g (group II) to 31.1 ng/g FW (group IV) during drop I (June drop). A significantly lower amount of ABA was found in the leaves of trees from cultivar groups I and II. ABA is a stress-regulated plant hormone, and 2015 was exceptionally dry. Therefore, significant differences in ABA in the cultivars from groups I and II, which build yield potential during blooming, may have been a result of water-deficit stress. The year 2016 was rainy, and the ABA amount in the leaves of all cultivars was similar and ranged from 23.4 to 29.4 ng/g FW at drop II (Table 4). In 2017, the precipitation amount was close to the multiannual average, and the amount of ABA ranged from 16.8 (group III) to 24.6 ng/g FW (group I). The significantly lowest ABA amounts at drop I were detected in the leaves of cultivars from groups III and IV (16.8 and 17.0 ng/g FW, respectively). We found that the fluctuations in ABA amounts were more related to environmental stresses experienced by plants, and we did not find trends between the studied cultivar groups.

Changes that occur in plant physiological processes are mostly determined by the phytohormone balance but not the amount of a certain plant hormone. The ratio of IAA/Z differed between the groups of cultivars in 2015. Group I cultivars had the highest ratio of IAA/Z at drops I and II, reaching 39.6 and 47.5, respectively. The lowest IAA/Z ratio (3.9) was detected in the cultivars of group IV. In 2016, the auxin to zeatin ratio was much lower in most of the cultivar groups; however, similar trends to 2015 were observed: the highest IAA/Z ratio was found in cultivars from groups I and II (18.2 and 12.2, respectively); the cultivars in these groups tend to have heavy blooming. The lowest IAA/Z ratio was identified in the leaves of cultivars from group IV, just 2.8. A large decrease in the IAA/Z ratio was observed in the leaves of cultivars of groups I and II at drop II in 2016. In 2017, the cultivars in groups I and II had a higher IAA/Z ratio. In general, the 3-year data on auxin to zeatin ratio showed that cultivars that bloom heavily (groups I and II) tend to have a higher IAA/Z ratio in their leaves during fruitlet drop, especially in unfavorable environmental conditions. In favorable conditions, the IAA/Z ratio decreases; however, a trend was observed. Such information can be used as a marker for the identification of heavy blooming seedlings in the early stages of breeding.

## 3. Discussion

Physiological fruitlet drop is common in all apple cultivars; however, in some cultivars, the rate of fruitlet drop is sufficient for good and qualitative yield; in many cases, it is too intense [17]. Apple varieties differ according to their fruiting habits, such as the natural early season abscission of blossoms and fruitlets. Only a small proportion of apple blossoms become fruitful: on average, only 5–10% of blossoms bear good-quality fruits [18,19]. Alburquerque et al. [20] reported that the intensity of blooming is determined by apple cultivar. The same tendency was established in this study, as the variation coefficient of blooming intensity in cultivars varied from 1.9% to 72.7% (Table 1). In our previous study, the cultivars were divided into five groups according to their blooming intensity and fruitlet self-elimination [21]. However, only four groups according to self-elimination habit were investigated (Table 1). The remaining fruitlet proportion ranged from 5.1% to 16.1% after drop II between cultivars from those groups. The natural abscission of fruitlets depends on genotype and environmental conditions, which may vary widely between different years. The self-regulation of apple fruit yield is determined by a large complex of traits. However, objective morphological, chemical, or molecular parameters that may allow the prediction of the self-elimination of fruitlet drop have not yet been developed [22]. Early identification and selection of apple genotypes with sufficient self-elimination of fruitlets are necessary and may be highly effective in reducing apple breeding process time [23].

The number of seeds in a fruit is one of the factors necessary for fruit development, as seeds may be involved in the production of phytohormones that regulate fruit nutrition and affect the shape and size of fruit [24]. Embryo abortion occurs when embryo development is disturbed, and seeds stop developing. Racskó et al. [16] found a direct correlation between fruit drop and the quantity of well-developed seeds in fruit: fruits with a lower number of well-developed seeds are dropped first. It has been established that fruits with less than three well-developed seeds are eliminated first [25]. The number of well-developed seeds varied according to blooming intensity and fruitlet self-elimination between 78% and 90.3% in the different cultivars groups (Figure 1). The number of developed seeds correlated with the number of remaining fruits, whereas the number of non-developed seeds had a negative correlation with fruit survival (Table 2). Pome fruit cultivars with a lower number of seeds are more susceptible to environmental conditions and tend to self-eliminate more fruits [26]. The share of developed seeds in the eliminated fruits of apple cultivars ranged from 9.1% to 26.0% (Figure 2). Cultivars belonging to groups I and III require a lower number of developing seeds for fruit survival.

The fruit number per tree depends on the amount of nutrients produced in the leaves close to inflorescence [27]. First-year shoots use the same nutrients necessary for fruit [28]. A low leaf/blossom ratio during blooming reduces fruiting and accelerates the self-elimination of fruitlets. This correlation was established between first-year shoot growth and fruitlet drop. Analysis of the photosynthetic parameters showed that cultivars belonging to the cultivar groups categorized according to their fruiting habit had from 3.1 to 20.4 leaves per fruit on first-year shoots and from 7.6 to 81.5 leaves per fruit on fruiting branches at the June drop (drop I) (Figure 3 and Figure 4). The leaf number per fruit was much higher in cultivars self-eliminating a low number of fruitlets. The same trend was observed after studying leaf area per fruit. A positive correlation between fruit survival and leaf number and area per fruit on fruiting branches was established, whereas a negative correlation was observed for first year shoots. Therefore, it appears that the first-year shoots compete for nutrients with fruitlets at the June drop. Soltész [26] reported that apple cultivars with vigorous shoots tend to self-eliminate more fruitlets than cultivars with weak vegetative growth. Significant changes were found in the carbohydrate content in cultivars that self-eliminate a large number of fruitlets (groups I and III) at the June drop compared to during blooming (Table 3).

Plant hormones regulate fruit development [29,30,31]. Auxin and cytokinin are the main plant hormones responsible for fruit growth [32]. It has been established that seeds produce plant hormones, especially auxin, gibberellin, and cytokinin, which stimulate the development of soft tissues and determine fruit size [24,33]. Apple fruitlet self-elimination depends on various factors. It was shown that periods of carbohydrate excess or deficit are related to the effectiveness of chemicals used for fruitlet thinning [10]. The influence of IAA, ABA, GA3, and Z was evaluated on yield self-regulation in apple, and we found that IAA, ABA, and the ratio between IAA and Z had the largest impact (Table 4). The IAA amount in leaves of cultivar groups with heavy blooming was higher than in cultivars with moderate blooming during the June drop. ABA is well-known as a stress hormone; the amount of ABA in cultivars with heavy blooming was lower during drought.

Physiological plant processes are controlled by plant hormones and their complexes. The cultivars that bloom heavily tend to have higher IAA/Z ratios in leaves during fruitlet drop compared to cultivars with moderate blooming. This ratio may be used to characterize cultivars’ fruiting habits. The established trends on the correlation of apple’s biological traits of blooming intensity and fruitlet self-elimination may be used in the breeding of cultivars with the desired fruiting pattern.

## 4. Materials and Methods

In total, 19 apple (*Malus × domestica* Borkh.) cultivars were selected for the study of yield self-regulation habits (Table 1). These cultivars were grouped into 4 groups according to long-term data on blooming and fruit-bearing [19]. The apple cultivars were planted with four replicates in one block per cultivar in the year 2004. Plants were grown on B118 rootstock, and trees were planted in a 5 × 3 m design. The experiment was conducted at the Lithuanian Research Centre for Agriculture and Forestry, Institute of Horticulture, Babtai Lithuania, located at 55°08′ N and 23°80′ E, at an elevation of 55 m. The climate is a humid continental type with an annual average precipitation of 630 mm and an average temperature of −5 °C in January and 17.3 °C in July.

### 4.1. Meteorological Conditions

The meteorological conditions during the experiment years (2015–2017) were considerably different (Appendix A). The year 2015 had periods of drought. Meteorological conditions were favorable during blooming in May. In June, during fruitlet set and June drop, a lack of humidity was observed, and rainfall was 48 mm lower than the multiannual average. The temperature was close to the multiannual average. In July, just before the second fruitlet drop, the weather was dry, and it remained dry in August, when only 6.9 mm of rainfall was observed compared to the multiannual average of 80.3 mm; the temperature was 4 °C higher than average.

The year 2016 was humid. In June, during blooming, the weather was warm and dry: there were 17 mm less rainfall and a 3.5 °C higher temperature compared to the multiannual averages. There was 21 mm more rainfall compared to the average during fruitlet set and June drop, and, in July, at the second fruitlet drop, rainfall was more than twice as high as usual, and the temperature was close to the multiannual average.

At the beginning of vegetation, in April of 2017, twice-higher rainfall (78 mm) was noticed; when dry weather prevailed during blooming in May, the temperature was slightly higher. June was humid, with 30 mm more rainfall than the multiannual average, and weather conditions in July were similar to the multiannual average.

### 4.2. Evaluation of Blooming Abundance

The abundance of blooming length of branch per blossom was evaluated during mass blooming. We analyzed 100 inflorescences on 3–4 year-old fruiting branches in 4 repetitions. Each repetition included 25 inflorescences per branch. Branches were located in different parts of the tree canopy at a 1.5–1.8 m height. Branch length from the apical bud to the basal part was measured; the number of blossoms was counted, and the average length of a branch per flower was calculated.

### 4.3. Evaluation of Dynamics of Blossom and Fruitlet Self-Elimination

The dynamics in blossom and fruitlet number were evaluated during vegetation; 100 inflorescences were selected at mass blooming (80% flowering buttons), and the initial blossom number was established. The remaining fruitlets were evaluated at the June drop (drop I) and in the middle of July (drop II); the share of eliminated fruitlets was calculated (as a percentage).

### 4.4. Assessment of Seed Development

After every fruit self-elimination period (drops I and II), 12 self-eliminated fruitlets and 12 still-growing fruits as control were picked (1 fruitlet–1 replication). Seeds were removed from the fruits and sorted into undeveloped (significantly smaller than others, collapsed, or only rudiments) and developed seeds characteristic of each stage of development. The number of developed and undeveloped seeds per fruit was counted. The minimum quantity of seeds required for fruit development in apple tree cultivars was calculated according to the minimal number of seeds available in developed fruits.

### 4.5. Assessment of Leaves Number and Area 

Leaves were collected from first-year shoots and bearing branches in June in 4 replicates. All leaves were collected from first- and third-year shoots (2–3 pcs.), including lateral branches to calculate the number of leaves per branch length unit (1 m). Leaves were counted, and their surface area was measured using the WinDias leaf image analysis system (Delta—T Devices Ltd., Burwell, UK). The number and area of leaves from first-year shoots per remaining fruitlet were calculated, and the same parameters were calculated per bearing branch. 

### 4.6. Phytohormone Analysis

Apple leaf samples were collected during fruitlet drop in June and July, labeled drop I and drop II, respectively. The sample combined 5 leaves collected from a well-illuminated part of the tree (the southern-facing external tree canopy) at a height of 1.5–1.8 m. Samples were immediately frozen and stored at −70 °C until phytohormone extraction.

Extraction was performed according to a modified version of Wang et al.’s [34] method. We ground 1 g of leaves in liquid nitrogen and resuspended it in 15 mL isopropanol (Sigma-Aldrich, Darmstadt, Germany). The samples were extracted for 24 h, then centrifuged. The supernatant was discarded, and the remaining pellet was resuspended in 2 mL of isopropanol. This step was repeated 3 times. Later extracts were cleaned using a rotary evaporator; phytohormones were isolated using SPE with NH_2_ (Supelco, Bellefonte, Pennsylvania, USA) and concentrated using a vacuum concentrator (Eppendorf Concentrator 5301, Eppendorf AG, Hamburg, Germany) to 0.5 mL. Samples were stored at 4 °C until analysis.

Plant hormones were determined using an HPLC (Agilent 1200, Agilent Technologies Inc., Waldbronn, Germany) system with a diode array detector. Plant hormones were separated on an Eclipse XDB C 18 Column (150 × 4.6 mm, size of particles = 3.5 μm) (Agilent Technologies Inc., Germany). Gradient elution was used; mobile phase A was 50% methanol (Sigma-Aldrich, Waldbronn, Germany), and B was 50% methanol (Sigma-Aldrich, Darmstadt, Germany) with 1.2% acetic acid (Sigma-Aldrich, Darmstadt, Germany). Gradient elution was performed as follows: 0 min 50% B, 3 min 50% B, 7 min 60% B, 8 min 50% B, and 10 min 50% B. Gibberellic acid 3 (GA3) and abscisic acid (ABA) were detected at 254 nm, zeatin (Z) at 270 nm, and indolyl-3-acetic acid (IAA) at 280 nm. Plant hormones were identified according to retention time and quantified using external standards of IAA, AGA, and GA3 (Duchefa Biochemie B.V., Haarlem, The Netherlands) and ABA (Alfa Aesar GMBH, Kandel, Germany).

### 4.7. Analysis of Soluble Carbohydrates 

Leaves were collected after blooming (as a control) and after June drop (II) in the year 2020. We ground 0.5 g of fresh plant leaves tissue and diluted them with 2 mL deionized H_2_O. The extraction was carried out for 4 h at room temperature with stirring. Samples were centrifuged at 14,000× *g* for 15 min. We mixed 1 mL of the supernatant with 1 mL 0.01% (*w*/*v*) ammonium acetate in acetonitrile and incubated for 30 min at 4 °C. The samples were filtered through a 0.22 μm PVDF filter (Carl ROTH GmbH + Co. KG., Karlsruhe, Germany). Carbohydrates were analyzed according to Ma et al. [35] with modifications. The analyses were performed on a Shimadzu HPLC (Shimadzu, Kyoto, Japan) instrument equipped with an evaporative light scattering detector (ELSD). Fructose, mannose, glucose, sucrose, maltose, and raffinose were separated on a Shodex VG-50 4D HPLC column with a deionized water (mobile phase A) and acetonitrile (mobile phase B) gradient. The gradient was maintained at 88% B for 13 min, changed linearly to 70% B in 9 min, kept at 70% B for 1 min, and increased back to 88% B for 2 min, and the column was equilibrated to 88% B for 5 min. The flow rate was 0.8 mL min^–1^. The experiment was replicated 3 times.

### 4.8. Statistical Analysis

Mean, standard deviation, and standard error of mean were calculated using MS Excel 2010. Significant differences were established using Duncan’s criteria by the dispersive ANOVA method in the *SELEKCIJA* software package [36]. Correlation and regression analysis was performed using STAT_ENG [36]; common tendencies for 3 years were established.

## 5. Conclusions

The apple cultivars differ in terms of the total number of seeds and the minimum number of seeds required for fruit development. We established that apple cultivars self-eliminating a small number of fruitlets need a lower number of well-developed seeds in fruit and their leaves number and area per fruit on fruiting branches is higher compared to cultivars self-eliminating large number of fruitlets. Our study shows that a higher carbohydrate amount is related to less fruitlet self-elimination. Phytohormones play an important role in the process of the elimination of fruitlets. The auxin amount and a high IAA/Z ratio in the leaves of cultivar groups with heavy blooming were higher than in cultivars with moderate blooming. All these features can be used as markers for the selection of different self-eliminating-fruitlets apple genotypes.

## Figures and Tables

**Figure 1 plants-10-01612-f001:**
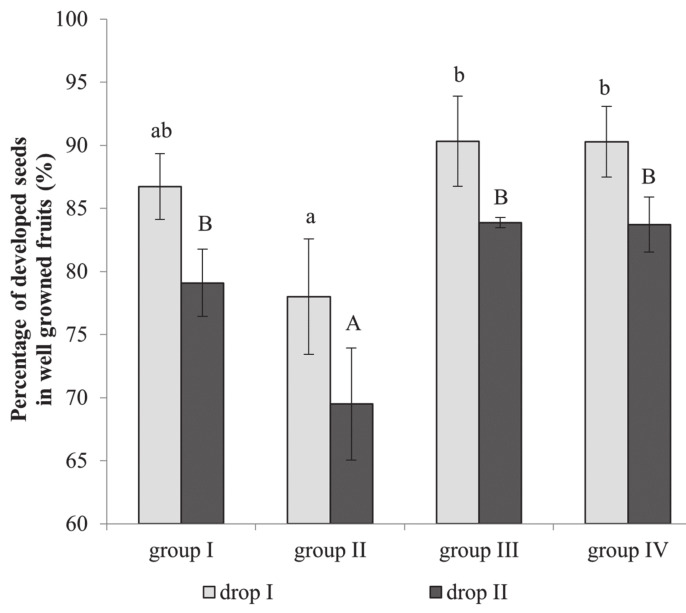
The percentage of developed seeds in fruits of various apple tree cultivars, grouped according to yield self-regulation habit. A 3-year average is presented. Bars indicate standard error of mean. Different letters indicate statistically significant differences: a,b: drop I; A, B: drop II (*p* ≤ 0.05).

**Figure 2 plants-10-01612-f002:**
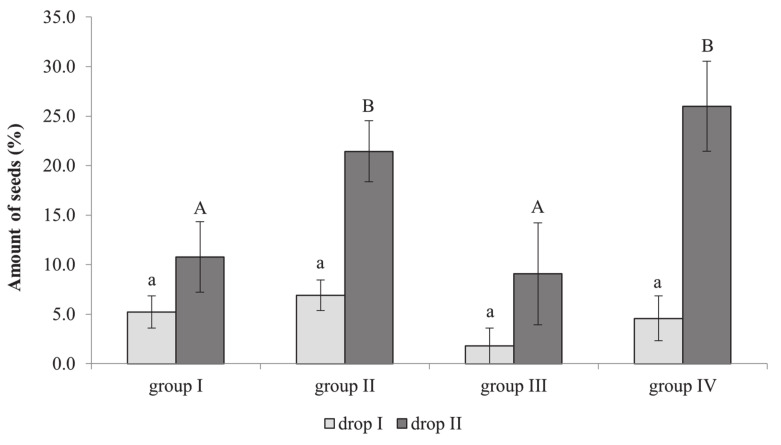
The percentage of developed seeds in eliminated immature fruits of various apple tree cultivars grouped according to their yield self-regulation habit. A 3-year average is presented. Bars indicate standard error of mean. Different letters indicate statistically significant differences: a, b: drop I; A, B: drop II (*p* ≤ 0.05).

**Figure 3 plants-10-01612-f003:**
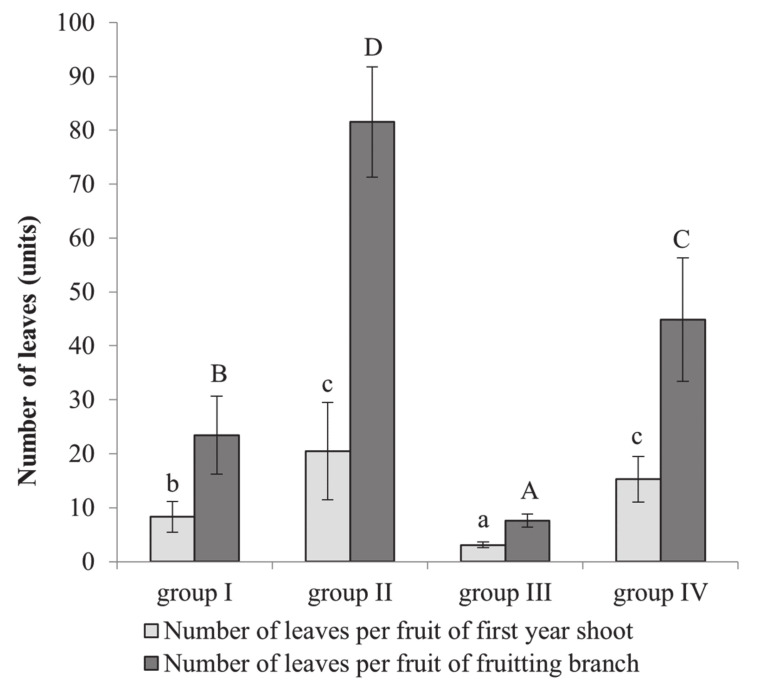
The number of leaves per fruit on first-year shoots and fruiting branches, in the different apple groups according to their yield self-regulation habits during the June-drop; a 3-year average is presented. Bars indicate the standard error of mean. Different letters indicate statistically significant differences: a, b, c: first-year shoots; A, B, C, D: fruiting branches (*p* ≤ 0.05).

**Figure 4 plants-10-01612-f004:**
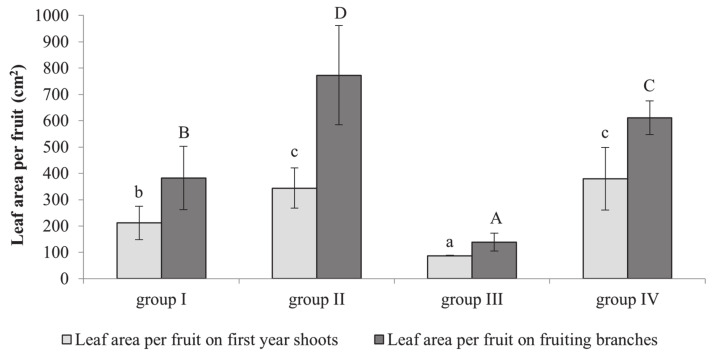
The leaf area per fruit on first year shoots and fruiting branches in different apple cultivar groups according to their yield self-regulation habit during the June drop; a 3-year average is presented. Bars indicate standard error of mean. Different letters indicate statistically significant differences: a,b,c: first-year shoots; A,B,C,D: fruiting branches (*p* ≤ 0.05).

**Table 1 plants-10-01612-t001:** Dynamics of the number of fruitlets during apple tree vegetation in 2015–2017.

Cultivar	3-Year Average Number of Blossoms per 1 m of Branch ± SD ^1^	Coefficient of Variation (%)	Drop I Number of Fruitlets per 1 m of Branch (%)			Drop II Number of Fruitlets per 1 m of Branch (%)		
			2015	2016	2017	2015	2016	2017
Orlovim	79.4 ± 18.3	23.1	38.4	21.4	8.5	19.1	10.9	6.2
Ottawa	49.9 ± 12.0	24.0	59.7	33.5	34.9	23.7	19.0	17.1
Sonata	63.7 ± 10.1	15.9	22.8	16.4	18.8	8.1	6.4	5.6
Lord Lemburne	77.3 ± 56.2	72.7	16.0	9.7	5.8	6.0	4.6	3.1
Melrose	52.8 ± 15.4	29.2	25.5	27.0	11.0	6.3	18.6	4.8
Selena	81.3 ± 15.5	19.0	61.8	47.7	28.7	20.6	20.0	12.9
Makresa	61.3 ± 21.8	35.6	28.8	31.4	26.5	9.9	17.7	7.9
Montvilinis	88.2 ± 4.7	5.3	46.8	61.4	19.5	20.0	23.7	12.8
McIntosh	52.7 ± 16.3	30.9	9.8	13.2	2.7	3.4	0.6	2.7
**Average of group I**	**67.4 ± 11.6**	**17.2**	**32.3**	**30.9**	**18.6**	**14.5**	**14.0**	**8.9**
Yellow Arkad	46.4 ± 15.1	32.5	36.9	20.6	16.3	23.3	13.6	11.4
Discovery	88.4 ± 36.3	41.0	10.7	3.2	3.2	8.7	1.9	2.3
Albrechtapfel	96.2 ± 35.3	36.7	30.8	5.1	3.3	20.5	3.8	2.2
Osvald	57.7 ± 1.3	2.3	25.9	9.6	21.3	10.2	7.8	9.9
Voshod	67.8 ± 1.3	1.9	0	0.7	-	0	0.3	0
**Average of group II**	**71.1 ± 14.1**	**19.9**	**20.1**	**7.7**	**8.9**	**12.8**	**5.4**	**5.1**
Port Oxford beauty	44.1 ± 16.6	37.6	29.3	54.0	43.5	15.1	13.2	14.5
Aldas	29.8 ± 13.1	43.9	40.8	32.1	36.4	16.7	17.2	30.9
Sandov	36.2 ± 4.2	11.5	54.7	66.2	52.9	17.2	24.3	20.0
**Average of group III**	**35.6 ± 5.3**	**14.8**	**49.6**	**50.8**	**38.8**	**20.2**	**18.5**	**16.3**
Wonchester permine	44.5 ± 13.9	31.3	16.2	10.4	7.3	12.3	5.8	5.9
Hume	32.2 ± 5.5	17.0	15.9	5.5	11.1	9.6	1.6	11.1
**Average of group IV**	**38.3 ± 9.3**	**24.2**	**16.0**	**8.4**	**8.6**	**10.8**	**4.1**	**7.8**

^1^ Standard deviation.

**Table 2 plants-10-01612-t002:** Correlation coefficients between seed development and fruit survival in the different apple tree cultivar groups.

	Group I		Group II		Group III		Group IV		All Groups	
	Drop I	Drop II	Drop I	Drop II	Drop I	Drop II	Drop I	Drop II	Drop I	Drop II
fruitlet survival	well-developed seeds									
	0.141	0.300	0.063	0.536 *	0.430	0.669 *	0.995 *	0.801 *	0.045	0.045
	non-developed seeds									
	0.045	-0.654*	0.141	−0.650 *	0.430	−0.879 *	−0.995 *	0.427 *	0.173	0.095

* Statistically significant at *p* ≤ 0.05.

**Table 3 plants-10-01612-t003:** Correlations between apple morphologic traits and leaf number and area on first-year shoots and fruiting branches.

	Group I	Group II	Group III	Group IV
Leaf number on first year shoot at drop I				
Tree load (blossoms)	−0.686 *	−0.732 *	−0.224	−0.913 *
Fruit survival, %	−0.825 *	−0.984 *	−0.224	−0.938 *
Number of developing seeds in fruit at drop I	0.371	0.666 *	0.766 *	0.970 *
Leaf number on fruiting branch at drop I				
Tree load (blossoms)	0.624 *	0.326	0.919 *	0.995 *
Fruit survival, %	0.618 *	0.998 *	0.928 *	0.792 *
Number of developing seeds in fruit at drop I	0.796 *	0.891 *	−0.063	−0.728 *
Leaf area on first year shoot at drop I				
Tree load (blossoms)	−0.598 *	−0.138	−0.599	−0.774 *
Fruit survival, %	−0.095	−0.976 *	−0.618	−0.314
Number of developing seeds in fruit at drop I	0.445	0.848 *	0.443	0.200
Leaf area on fruiting branch at drop I				
Tree load (blossoms)	0.814 *	0.451	0.445	1.000 *
Fruit survival, %	0.624 *	0.929 *	0.423	0.807 *
Number of developing seeds in fruit at drop I	0.363	0.300	0.999 *	0.861 *
Number of developing seeds in fruit at drop II	0.956 *	−0.373	0.966 *	−1.000 *

* Statistically significant at *p* ≤ 0.05.

**Table 4 plants-10-01612-t004:** Changes in carbohydrate amount in apple leaves at drop II compared to blooming.

	Fructose		Sorbitol		Glucose		Total	
Cultivar	Difference mg/g	Amount (%)	Difference mg/g	Amount (%)	Difference mg/g	Amount (%)	Difference mg/g	Amount (%)
Orlovim	−3.04 *	71.2	−12.93 *	76.9	−5.78	83.3	−21.76 *	78.5
Otava	−2.68 *	80.2	−19.63 *	71.6	−6.14	83.1	−28.45 *	76.1
Sonata	−2.75 *	70.4	−7.54 *	82.4	−15.56 *	62.4	−25.84 *	62.4
Lord Lembourne	0.11	101.2	−4.69	89.9	−1.37	94.4	−5.95	94.4
Melrose	−1.51	84.0	−12.29 *	80.0	−3.23	92.4	−17.03 *	85.0
Selena	−1.99	79.0	−9.97 *	81.4	−0.68	98.0	−12.64	87.0
Mackresa	0.42	103.8	−0.02	99.6	−9.19 *	75.8	−8.97	90.5
McIntosh	1.59	128.1	6.78	118.5	4.67	119.7	13.04	119.8
Average of group I	−1.23	89.7	−7.54 *	87.5	−4.66	88.6	−13.45 *	86.7
Yellow Arkad	1.98	135.1	−0.56	98.6	−12.93 *	70.8	−11.52	87.2
Discover	1.35	114.2	5.89	113.0	1.38	104.4	8.62	110.0
Albrechtapfel	−1.65	84.9	−4.28	89.4	−13.11 *	67.9	−19.04 *	79.4
Osvald	−1.70	85.9	−6.20	88.1	−6.14	82.4	−14.04 *	82.4
Voshad	−1.63	80.4	10.73 *	134.1	−6.94	81.9	2.16	102.8
Average of group II	−0.33	100.1	1.12	104.6	−7.55	81.5	−6.76	92.7
Aldas	−3.59 *	68.2	−14.76 *	72.8	−4.00	91.6	−22.34 *	80.2
Sandow	0.23	102.7	−20.88 *	65.3	−17.27 *	58.3	−37.92 *	65.6
Average of group III	−1.68	85.5	17.82 *	69.1	−10.64 *	75.0	−30.13 *	72.9
Worchester Permain	−2.50 *	78.1	−2.18	95.4	−13.83 *	69.5	−18.52 *	82.2
Hume	0.59	106.3	2.46	105.1	5.01	116.9	8.06	109.3
Average of group IV	−0.96	92.2	0.14	100.3	−4.41	93.2	−5.23	95.8
LSD 0.05	2.49		7.23		7.60		13.16	

* Statistically significant at *p* ≤ 0.05.

**Table 5 plants-10-01612-t005:** Plant hormones in the leaves of various apple cultivars.

Plant Hormone	Group I ^1,^^2^	Group II	Group III	Group IV
**IAA ng/g FW**				
2015 drop I	274.7 ± 50.0 ab	358.1 ± 50.0 b	224.6 ± 67.1 a	168.0 ± 39.1 a
2016 drop I	715.5 ± 73.6 d	490.0 ± 70.8 c	320.3 ± 82.9 b	121.3 ± 48.5 a
2017 drop I	213.3 ± 55.6 c	200.2 ± 44.1 c	88.9 ± 9.2 a	132.9 ± 28.7 b
**Average drop I**	**401.2 ± 31.3 c**	**349.4 ± 20.7 c**	**211.3 ± 33.0 b**	**140.7 ± 6.4 a**
2015 drop II	305.3 ± 57.9 B	351.2 ± 60.0 B	277.6 ± 65.0 AB	192.9 ± 13.4A
2016 drop II	148.8 ± 15.7 A	241.1 ± 40.2 B	274.12 ± 53.4 B	155.3 ± 19.2 A
2017 drop II	262.5 ± 66.4 B	362.8 ± 50.3 B	91.5 ± 12.8 A	319.2 ± 14.4 B
** Average drop II**	**238.9 ± 29.2 A**	**318.4 ± 27.8 B**	**214.4 ± 18.3 A**	**222.5 ± 6.8 A**
**Z ng/g FW**				
2015 drop I	6.9 ± 0.6 a	18.6 ± 9.5 b	47.3 ± 10.5 c	46.5 ± 9.8 c
2016 drop I	42.0 ± 5.1 ab	39.0 ± 2.3 a	55.4 ± 10.1 b	41.3 ± 7.3 ab
2017 drop I	56.0 ± 3.2 b	57.2 ± 5.0 b	48.5 ± 3.4 a	45.5 ± 4.4 a
**Average drop I**	**34.9 ± 1.1 a**	**38.3 ± 1.2a**	**50.4 ± 8.6 b**	**44.4 ± 2.3 ab**
2015 drop II	6.8 ± 0.62 A	22.6 ± 11.4 B	29.5 ± 9.7 B	46.3 ± 4.5 C
2016 drop II	36.6 ± 2.1 **A**	73.9 ± 6.2 C	45.7 ± 3.8 B	31.2 ± 3.4A
2017 drop II	51.1 ± 4.7 AB	55.0 ± 3.0 B	48.2 ± 1.4 A	49.3 ± 2.4 A
** Average drop II**	**31.5 ± 2.3 A**	**59.7 ± 7.1 C**	**41.2 ± 5.7B**	**42.2 ± 0.8 B**
**GA3 µg/g FW**				
2015 drop I	1.6 ± 0.1 b	3.0 ± 0.2 c	1.2 ± 0.1 a	1.1 ± 0.1 a
2016 drop I	2.3 ± 0.1 a	2.5 ± 0.3 a	1.9 ± 0.4 a	2.3 ± 0.5 a
2017 drop I	1.0 ± 0.3 b	1.4 ± 0.3 b	0.6 ± 0.05 a	0.45 ± 0.05 a
**Average drop I**	**1.7 ± 0.2 ab**	**2.3 ± 0.2 b**	**1.2 ± 0.2 a**	**1.3 ± 0.2 a**
2015 drop II	1.5 ± 0.1 A	2.8 ± 0.2C	2.2± 0.2 B	2.6 ± 0.3C
2016 drop II	2.6 ± 0.4 A	2.5 ± 0.4 A	3.2 ± 0.1 B	2.0 ± 0.7 A
2017 drop II	0.6 ± 0.1 A	1.4 ± 0.3 B	0.6 ± 0.4 A	1.0 ± 0.1 AB
** Average drop II**	**1.6 ± 0.2 A**	**2.2 ± 0.3 B**	**2.0 ± 0.2 AB**	**2.0 ± 0.3 AB**
**ABA ng/g FW**				
2015 drop I	14.0 ± 1.6 a	10.6 ± 1.5 a	20.5 ± 2.0 b	31.1 ± 2.5 c
2016 drop I	23.4 ± 2.5 a	29.4 ± 2.1 b	29.1 ± 3.2 ab	26.4 ± 5.2 ab
2017 drop I	24.6 ± 7.9 a	27.4 ± 6.6 a	20.7 ± 8.8 a	20.4 ± 7.2 a
**Average drop I**	**20.7 ± 3.6 a**	**22.5 ± 2.8 ab**	**23.4 ± 2.2 ab**	**25.9 ± 0.7 b**
2015 drop II	17.7 ± 3.4 A	12.3 ± 3.0 A	28.0 ± 2.5 B	30.6 ± 3.0 B
2016 drop II	32.1 ± 0.6 A	37.9 ± 2.4 B	41.6 ± 2.0 C	30.4 ± 1.2 A
2017 drop II	24.2 ± 5.7 B	22.7 ± 6.0 AB	16.8 ± 0.6 A	17.0 ± 1.1 A
** Average drop II**	**24.7 ± 2.9 A**	**24.3 ± 1.7 A**	**28.8 ± 2.3 A**	**26.0 ± 1.8 A**
**IAA/Z ratio**				
2015 drop I	39.6 ± 6.9 c	28.0 ± 3.3 b	22.2 ± 4.0 b	3.9 ± 1.5 a
2016 drop I	18.2 ± 2.3 d	12.2 ± 1.9 c	7.6 ± 2.1 b	2.8 ± 0.7 a
2017 drop I	3.7 ± 0.8 a	3.7 ± 0.9 a	2.0 ± 0.5 a	2.9 ± 0.1 a
**Average drop I**	**20.5 ± 3.0 c**	**14.6 ± 2.3 b**	**10.6 ± 4.5 b**	**3.2 ± 0.3 a**
2015 drop II	47.5 ± 7.1 C	16.6 ± 2.3B	17.3 ± 5.8 B	4.2 ± 0.7 A
2016 drop II	4.1 ± 0.4 B	4.4 ± 1.0 A	6.1 ± 1.7 AB	5.1 ± 1.2 A
2017 drop II	5.4 ± 0.8 B	6.2 ± 2.0 B	1.9 ± 0.2 A	6.5 ± 0.8 B
** Average drop II**	**19.0 ± 3.6 B**	**7.8 ± 2.2 A**	**8.42 ± 3.9 A**	**5.3 ± 0.4 A**

^1^ Standard deviation, ^2^ different letters indicate statistically significant differences between the four groups of cultivars: a, b, c, d: drop I; A, B, C: drop II (*p* ≤ 0.05).

## Data Availability

Data available in a publicly accessible repository. The data presented in this study are openly available in [repository name e.g., FigShare] at [doi], reference number [reference number].

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
