# Peer review of "Potential Markers for Selecting Self-Eliminating Apple Genotypes"

_plants, 2021, doi:10.3390/plants10081612_

Round 1
Reviewer 1 Report
I have read the article entitled "Potential markers for selecting fruitlets self-eliminating apple genotypes", submitted by Starkus et al. for consideration for publication in Plants. This article investigates the relative importance of fruitlets dropping on determining patterns of fruit set in several varieties of apples in Lithuania over different fruiting seasons.
The general idea of the article is interesting and promising and the data set seems to be quite expressive. However, I found it difficult to clearly understand the main picture of the manuscript. The text is massively descriptive with several levels of detail making it very hard to digest - maybe it is my lack of ability to follow it. This was probably my major concern, which affects the understanding of the entire article. I think that most of it would be solved if the authors present a clear objective with hypotheses and predictions. It is not strictly clear what is being tested and why, and many relations just appeared in the results. The statistical analysis section is very vague and do let clear the models tested.
I would recommend the authors to rethink the entire article presentation in order to make the results section more cohesive and attractive. I point below some specific comments that might help to improve the MS.
Based on my detailed reading, I think that this article would be of interest to the Plants readers, however, it still demands substantial work and new evaluation before being considered for publication. Therefore, my recommendation is major revision.
Specific comments:
Line 34: Is this true worldwide? All cultivars? If so, please provide a reference for this.
Line 70-71: I found this objective paragraph overly succinct. I think that the authors could elaborate it a bit more, perhaps by adding some questions and predictions.
Lines 341-357: All this information seems relevant but also dense, perhaps a supplementary figure showing all this variation would be helpful here.
Lines 377 and 378: Please, be consistent with the space or not after the number.
Line 377: Why one meter? Is there any biological reason for this?
Line 385: Please explain what is "well illuminated part of the tree" as it can be quite subjective. Give some reference, so any reader interested in repeating your experiments could make it.
Line 386: Hormone extraction?
Line 386: I am not sure if I understand correctly "After extraction for 24 hours", could you please explain it better or check the English.
Lines 420-424: Unfortunately, your sample design is not clear so I can't offer a more deep advice regarding statistical analysis. I would strongly encourage the authors to provide in detail all relations tested. What were your dependent and independent variables in your model? What is your model? Perhaps this issue is related to lack of clear objectives as I pointed above.
Line 77: I can't see any inferential test for such non-significant differences.
Line 100: You don't need to show all the Y-axis range in this plot, limiting the range between 60 to 100 would reduce the amount of used space and also evidence the differences. Further, I would recommend using more neutral colors to fill the bars and a black contour line to delimit it. This applies to all plots.
Line 120: I think that "significant differences" is being used improperly, as you don't provide the inferential tests parameters for this.
Lines 119-131: All this paragraph is a bit hard to follow, perhaps some figures and tables would help on the understanding.
Lines 143-145: How did you reach such a conclusion?
Line 69: What type of correlation? Pearson?
Line 172: please check the plot y-axis label, it is not centered and with a comma. General comment for all plots: if the comparison is made within color (lower- and upper-case letter upon the bars), you should group the bars by colors, being one category of leaf area per fruit on first year shoots with four bars side by side, and the other category about fruiting branches with four bars.
Line 202: Please check the average numbers by group as there are some inconsistencies. For instance, the "Fructose difference mg/g" of the "Average of group I" is set as 1.23, but calculating here the average is -0.75, the "Amount (%)" is also wrong. Please check it or explain why such inconsistencies exist.
Line 259: Why not present plots instead of this table? The table is hard to follow and find clear patterns.
Author Response
We are thankful for the valuable comments of the reviewer which are helpful in our commitment to produce a high-quality manuscript. Changes in the manuscript (with track changes in the manuscript) were made according to the reviewer comments. Please find in attached word documents answers to all your questions and suggestions.
Reviewer 2 Report
The purpose of this manuscript was to identify factors important for limiting mature fruit formation in apple trees via dropping of immature fruits (fruitlets). The authors used 19 different cultivars of apple in this study and tracked blossom number, fruitlet drop at two time points, seed set, leaf abundance, leaf area, leaf carbohydrates, and leaf hormones. This work has the potential to uncover some of the parameters of influence for the trait of interest. However, some concerns remain.
Major concerns
- The experimental design must be better described. How many trees were planted of each cultivar? Were trees randomized? How many planting blocks were present? Were border trees included in the analysis? What was the age of the grafted trees at planting?
- The grouping of plants into 5 categories by 1 trait (bloom load) is artificial and may actually be masking results of interest. It would be more biologically meaningful to keep each cultivar as an individual and let the data analysis determine which trait(s) are predictive of fruitlet drop and fruit set. Please re-analyze the data using cultivar as a term and see how the data group. Principle components analysis (PCA) would be one way to examine the data, or ANOVA to see if cultivar is a significant factor (or to see if the groups are of significance).
- The methods need clarification. For example, were the blooming abundances counted on 4 different trees (the repetitions) or the same tree? Similarly, were the 12 fruits and 12 fruitlets used for seed development from the same tree or different trees? Were the replicates used for leaf area the same tree or different trees? Please clarify for all experiments.
- What is being shown in Table 2? The title is about leaf number and leaf area, but the row titles are of tree load and fruit survival. Please clarify and edit as needed.
- Table 3 is lacking the yearly data, it shows the change in carbohydrate levels but not the actual yearly levels. It would be very informative to show all the data. Please add them (or include as a supplementary file).
- Please add additional information on the desired abundance of fruits per tree and how much fruitlet drop would be wanted. This could be descriptive, not necessarily numeric.
- The data interpretation was a bit confusing, please clarify the following information. In table 1, Drop 1 and Drop 2 is “number of fruitlets per 1 m of branch %.” Are the data shown a number or a percentage? If this is a %, how was it calculated? How was seed number calculated? According to page 3 line 90 there were “15.5 and 13.6” seeds in two of the cultivars. However, a standard apple forms just 10 total seeds per fruit. Is this total seeds for the 12 fruits examined?
- Did the cultivars studied show evidence of bi-annual bearing? Please clarify.
Minor concerns
- Please include a supplementary file of all data and of any coding used for data analysis.
- The abstract and first paragraph of the introduction are highly similar. Please revise the writing to make them more distinct from each other.
Author Response
We are thankful for the valuable comments of the reviewer which are helpful in our commitment to produce a high-quality manuscript. Changes in the manuscript (with track changes in the manuscript) were made according to the reviewer comments. Please find in attached word documents answers to all your questions and suggestions

Reviewer 3 Report
The presented research in the manuscript is in line with the methodology of creating scientific papers. English grammar proofreading required. The summary should emphasize the importance of the research carried out. The materials and methods section should be more precise. Not everything is clear and transparent. The materials and methods section should be placed after the introduction section. The presented conclusions are summaries, not conclusions.
Author Response

(The authors gave the same response as above.)

Round 2
Reviewer 1 Report
I think that the authors did a nice job by restructuring the article according to most of my recommendations, but I still think that the English could be improved, which will probably done at MDPI.
Therefore, based on may reading, I am happy to recommend this paper for publication in Plants.
Author Response
Thank you for the comment. As we noted it in the first round of review, we have sent our manuscript already to MDPI English service. So I hope you got that updated version of the manuscript.
Reviewer 2 Report
Thank you for the extensive revisions to your manuscript, the methods are clearer, and the paper is much improved by these refinements. It looks like you have a nice set of data and the potential for future research on some of the aspects.
A few comments remain
- The clarifications on the numbers of trees and tree age needs to go in the methods section, not just in the response to reviews. Please add this information. Also, how are the trees arranged in the field? Are the trees randomized or planted in blocks or planted in randomized blocks? Location can greatly influence tree performance. It was good to see that border trees were not included in the investigation.
- It is highly suggested to put your actual data (all your data, raw data, calculated data, R code) into the supplemental file. Right now the supplemental file is just the weather data for the locations which, while informative, is not key for your manuscript. The purpose of including all that data is two-fold. First, it allows others access to your data. Second, it means that all your data are in a centralized accessible location for your group. That way, none if it is lost or forgotten as researchers graduate or leave for other positions etc.
- This reviewer still feels it would be more informative to analyze cultivars as individuals, rather than in groups. However, this could be the starting point of some future work.
This last comment here is not meant to be included in the paper but instead is based on reviewer thoughts. Feel free to ignore this commentary as your group already has plans for future work. Do tree size and age play a factor in fruit drop or it that feature consistent across tree age? How much shared genetic background is there in each group? What is the actual fruit yield (apples per tree) for each cultivar in the absence of induced thinning (by chemicals or by hand)? How predictive are ABA levels on fruit drop? Out of curiosity, do the cultivars with more than 10 seeds per fruit show any floral alterations?
Author Response
Thank you for all comments, our answers are attached